# Interactive Actions of Aldosterone and Insulin on Epithelial Na^+^ Channel Trafficking

**DOI:** 10.3390/ijms21103407

**Published:** 2020-05-12

**Authors:** Rie Marunaka, Yoshinori Marunaka

**Affiliations:** 1Research Institute for Clinical Physiology, Kyoto Industrial Health Association, Kyoto 604-8472, Japan; marunakarie@gmail.com; 2Okamura Dental Clinic, Chuo-ku, Osaka 541-0041, Japan; 3Research Center for Drug Discovery and Pharmaceutical Development Science, Research Organization of Science and Technology, Ritsumeikan University, Kusatsu 525-8577, Japan; 4Department of Molecular Cell Physiology, Kyoto Prefectural University of Medicine Graduate School of Medical Science, Kyoto 602-8566, Japan

**Keywords:** ENaC, transcellular Na^+^ reabsorption, aldosterone, insulin, mathematical model, simulation, epithelium

## Abstract

Epithelial Na^+^ channel (ENaC) participates in renal epithelial Na^+^ reabsorption, controlling blood pressure. Aldosterone and insulin elevate blood pressure by increasing the ENaC-mediated Na^+^ reabsorption. However, little information is available on the interactive action of aldosterone and insulin on the ENaC-mediated Na^+^ reabsorption. In the present study, we tried to clarify if insulin would modify the aldosterone action on the ENaC-mediated Na^+^ reabsorption from a viewpoint of intracellular ENaC trafficking. We measured the ENaC-mediated Na^+^ transport as short-circuit currents using a four-state mathematical ENaC trafficking model in renal A6 epithelial cells with or without aldosterone treatment under the insulin-stimulated and -unstimulated conditions. We found that: (A) under the insulin-stimulated condition, aldosterone treatment (1 µM for 20 h) significantly elevated the ENaC insertion rate to the apical membrane ($k_{I}$) 3.3-fold and the ENaC recycling rate ($k_{R}$) 2.0-fold, but diminished the ENaC degradation rate ($k_{D}$) 0.7-fold without any significant effect on the ENaC endocytotic rate ($k_{E}$); (B) under the insulin-unstimulated condition, aldosterone treatment decreased $k_{E}$ 0.5-fold and increased $k_{R}$ 1.4-fold, without any significant effect on $k_{I}$ or $k_{D}$. Thus, the present study indicates that: (1) insulin masks the well-known inhibitory action of aldosterone on the ENaC endocytotic rate; (2) insulin induces a stimulatory action of aldosterone on ENaC apical insertion and an inhibitory action of aldosterone on ENaC degradation; (3) insulin enhances the aldosterone action on ENaC recycling; (4) insulin has a more effective action on diminution of ENaC endocytosis than aldosterone.

## 1. Introduction

The transepithelial Na^+^ transport mediated via epithelial Na^+^ channel (ENaC) participates in various bodily functions including regulation of blood pressure, the amount of body fluid content, and the lung alveolar fluid clearance [1,2,3,4,5,6,7,8,9,10,11,12,13]. This ENaC-mediated transepithelial Na^+^ transport requires two steps across the apical and basolateral membranes of epithelial cells: (1) the first step is the entry of Na^+^ across the apical membrane of epithelial cells into the intracellular space via ENaC expressed in the apical membrane, and (2) the second step is the extrusion of Na^+^ from the intracellular space across the basolateral membrane of epithelial cells mediated by the Na^+^,K^+^-pump located in the basolateral membrane [8,9]. It is generally considered that the rate-limiting step in the ENaC-mediated transepithelial Na^+^ transport is the Na^+^ entry through the apical-membrane-located ENaC rather than the Na^+^ extrusion mediated by the basolateral-membrane-located Na^+^,K^+^-pump [14]. This means that the amount of ENaC-mediated transepithelial Na^+^ transport mainly depends on the amount of the apical-membrane-located ENaC-mediated Na^+^ entry, which is determined by the number of ENaC and the activity (open probability; Po) of individual ENaC located in the apical membrane, and the driving force of Na^+^ entry across the apical membrane [9,15,16,17,18,19].

Aldosterone and insulin are well known to modulate the intracellular trafficking process of ENaC [20,21,22]. However, it is still unclear how aldosterone interacts with insulin in regulation of the intracellular ENaC trafficking process. We have recently established a mathematical model simulating the intracellular ENaC trafficking process [23,24]. Therefore, in the present study, we tried to clarify if insulin would modulate the effect of aldosterone on the insertion, endocytotic, recycling, and degradation rates of ENaC using the established four-state mathematical model of intracellular ENaC trafficking [23,24]. We here report for the first time the interactive action of aldosterone and insulin on the intracellular ENaC trafficking process: (1) insulin masks the well-known inhibitory action of aldosterone on the ENaC endocytotic rate; (2) insulin induces a stimulatory action of aldosterone on ENaC apical insertion and an inhibitory action of aldosterone on ENaC degradation; (3) insulin enhances the aldosterone action on ENaC recycling; and (4) insulin has a more effective action on diminution of ENaC endocytosis than aldosterone.

## 2. Results

### 2.1. Mathematical Model of Intracellular ENaC Trafficking

Figure 1 shows a mathematical model of ENaC trafficking consisting of four states: (1) an insertion state (Insert), (2) an apical membrane state (Apical), (3) a recycling state (Recycl), and (4) a degradation state (Degrad). ENaC in an insertion state (Insert) is trafficked to the apical membrane with the insertion rate, $k_{I}$. ENaC in an apical membrane state (Apical) can conduct Na^+^ and is retrieved to a recycling state (Recycl) with the endocytotic rate, $k_{E}$. ENaC in a recycling state (Recycl) is retrieved from an apical membrane state (Apical) with the endocytotic rate, $k_{E}$, and then trafficked to the insert state (Insert) with the recycling rate, $k_{R}$, communicating with the apical membrane state (Apical), or to a degradation state (Degrad) with the degradation rate, $k_{D}$ (see Figure 1).

The following four differential equations (see Equations (1)–(4)) respectively show the amounts of ENaC in (1) the insertion state (Insert), (2) the apical membrane state (Apical), (3) the recycling state (Recycl) and (4) the degradation state (Degrad):(1)$$\frac{d\;Insert\left(t\right)}{dt}=-k_{I}\;Insert\left(t\right)+k_{R}\;Recycl\left(t\right)$$
(2)$$\frac{d\;Apical\left(t\right)}{dt}=k_{I}\;Insert\left(t\right)-k_{E}\;Apical\left(t\right)$$
(3)$$\frac{d\;Recycl\left(t\right)}{dt}=k_{E}\;Apical\left(t\right)-\left(k_{R}+k_{D}\right)\;Recyc\left(t\right)$$
(4)$$\frac{d\;Degrad\left(t\right)}{dt}=k_{D}\;Recycl\left(t\right)$$
where t is the time after application of insulin (100 nM) or an insulin-solvent solution (water) alone to the fluid facing the basolateral membrane of cells with or without treatment of aldosterone (1 μM) for 20 h; Insert (t) means the amount of ENaC in the insertion state (Insert) at time = t; Apical(t) means the amount of ENaC in the apical membrane state (Apical) at time = t; Recycl(t) means the amount of ENaC in the recycling state (Recycl) at time = t; Degrad(t) means the amount of ENaC in the degradation state (Degrad) at time = t. $Insert_{0}$, $Apical_{0}$, $Recycl_{0}$, and $Dergad_{0}$ are respectively defined as the amounts of ENaC in states of Insert, Apical, Recycl and Degrad just before application of insulin or an insulin-solvent solution (water) (i.e., $Insert_{0}=Insert\left(0\right)$, $Apical_{0}=Apical\left(0\right)$, $Recycl_{0}=Recycl\left(0\right)$ and $Dergad_{0}=Degrad\left(0\right)$).

Equations (5)–(8) are respectively general solutions for Insert (t) (Equation (1)), Apical(t) (Equation (2)), Recycl(t) (Equation (3)) and Degrad(t) (Equation (4)).
(5)$$Insert\;\left(t\right)=C_{1}\frac{k_{E}+l}{k_{I}}exp\left(lt\right)+C_{2}\frac{k_{E}+m}{k_{I}}exp\left(mt\right)+C_{3}\frac{k_{E}+n}{k_{I}}exp\left(nt\right)$$
(6)$$Apical\;\left(t\right)=C_{1}exp\left(lt\right)+C_{2}exp\left(mt\right)+C_{3}exp\left(nt\right)$$
(7)$$Recycl\;\left(t\right)=C_{1}\frac{\left(k_{I}+l\right)\left(k_{E}+l\right)}{k_{I}\;k_{R}}exp\left(lt\right)+C_{2}\frac{\left(k_{I}+m\right)\left(k_{E}+m\right)}{k_{I}\;k_{R}}exp\left(mt\right)+C_{3}\frac{\left(k_{I}+n\right)\left(k_{E}+n\right)}{k_{I}\;k_{R}}exp\left(nt\right)$$
(8)$$Degrad\;\left(t\right)=C_{1}\frac{k_{D}\left(k_{I}+l\right)\left(k_{E}+l\right)}{k_{I}\;k_{R}\;l}exp\left(lt\right)+C_{2}\frac{k_{D}\left(k_{I}+m\right)\left(k_{E}+m\right)}{k_{I}\;k_{R}\;m}exp\left(mt\right)+C_{3}\frac{k_{D}\left(k_{I}+n\right)\left(k_{E}+n\right)}{k_{I}\;k_{R}\;n\;}exp\left(nt\right)+C_{4}$$
where $C_{1}$, $C_{2}$, $C_{3}$ and $C_{4}$ appearing in Equations (5)–(8) are respectively represented by Equations (9)–(12), while l, m and n appearing in Equations (9)–(12) are respectively one of the three roots, r, of the cubic Equation (13) (c.f., Cardano’s Formula for cubic equation).
(9)$$C_{1}=\frac{k_{I}\left(k_{I}+k_{E}+m+n\right)\;Insert_{0}-\left(k_{E}+m\right)\left(k_{E}+n\right)\;Apical_{0}-k_{I}k_{R}\;Recycl_{0}}{\left(l-m\right)\left(n-l\right)}$$
(10)$$C_{2}=\frac{k_{I}\left(k_{I}+k_{E}+l+n\right)\;Inseet_{0}-\left(k_{E}+l\right)\left(k_{E}+n\right)\;Apical_{0}-k_{I}k_{R}\;Recycl_{0}}{\left(l-m\right)\left(m-n\right)}$$
(11)$$C_{3}=\frac{k_{I}\left(k_{I}+k_{E}+l+m\right)\;Insert_{0}-\left(k_{E}+l\right)\left(k_{E}+m\right)\;Apical_{0}-k_{I}k_{R}\;Recycl_{0}}{\left(m-n\right)\left(n-l\right)}$$
(12)$$C_{4}=Degrad_{0}+\frac{k_{D}}{k_{R}\;l\;m\;n}\left\{\left[k_{I}k_{E}\left(k_{I}+k_{E}+l+m+n\right)-lmn\right]\;Insert_{0}-\left(k_{E}+l\right)\left(k_{E}+m\right)\left(k_{E}+n\right)\;Apical_{0}-k_{I}k_{E}k_{R}\;Recycl_{0}\right\}$$
(13)$$r^{3}+\left(k_{I}+k_{E}+k_{R}+k_{D}\right)r^{2}+\left(k_{I}k_{E}+k_{I}k_{R}+k_{I}k_{D}+k_{E}k_{R}+k_{E}k_{D}\right)r+k_{I}k_{E}k_{D}=0$$

### 2.2. The Short-Circuit Current (I_SC_) Was Sensitive to 10 µM Benzamil Under the Insulin-Stimulated and -Unsitmulated Conditions in Cells Treated with and without 1 µM Aldosterone for 20 h

To study if the I_SC_ observed in the present study would be the ENaC-mediated Na^+^ current, we applied 10 µM benzamil to the solution facing to the apical membrane where ENaC is localized, since benzamil is a specific inhibitor of ENaC at the concentration of 10 µM [23,25,26]. In cells without aldosterone treatment, apical addition of 10 µM benzamil abolished I_SC_ to 0.03 ± 0.01 µA/cm^2^ from 0.55 ± 0.05 µA/cm^2^ (*n* = 3; *p* < 0.001); 95% I_SC_ was sensitive to 10 µM benzamil. After addition of 10 µM benzamil, basolateral application of 100 nM insulin did not show any significant action on I_SC_ (0.04 ± 0.01 µA/cm^2^ 30 min after insulin application (*n* = 3); 0.05 ± 0.01 µA/cm^2^ 60 min after insulin application (*n* = 3)). In cells treated with 1 µM aldosterone for 20 h, apical addition of 10 µM benzamil abolished I_SC_ to 0.12 ± 0.02 µA/cm^2^ from 3.74 ± 0.28 µA/cm^2^ (*n* = 3; *p* < 0.001); 97% I_SC_ was sensitive to 10 µM benzamil. After addition of 10 µM benzamil, basolateral application of 100 nM insulin did not show any significant action on I_SC_ (0.15 ± 0.02 µA/cm^2^ 30 min after insulin application (*n* = 3); 0.09 ± 0.02 µA/cm^2^ 60 min after insulin application (*n* = 3)). Apical addition of 10 µM benzamil 5 h after insulin application respectively diminished the I_SC_ to 0.01 ± 0.01 µA/cm^2^ from 0.39 ± 0.06 µA/cm^2^ (*n* = 6; *p* < 0.001) in aldosterone-untreated cells and 0.06 ± 0.02 µA/cm^2^ from 2.20 ± 0.03 µA/cm^2^ (*n* = 6; *p* < 0.001) in aldosterone-treated cells.

These observations clearly indicate that most of I_SC_ measured in the present study was mediated via ENaC under both conditions with and without insulin stimulation irrespective of aldosterone treatment. Therefore, in the present study, we analyzed the measured I_SC_ considering that both the basal I_SC_ and the insulin-stimulated I_SC_ were the ENaC-mediated Na^+^ currents in cells with and without aldosterone treatment. Further, the insulin action on the ENaC-mediated I_SC_ is mainly mediated through stimulation of ENaC translocation to the apical membrane [27], although insulin has some minor action of the ENaC activity (the open probability (Po) of individual ENaC) [28].

### 2.3. The Time Course of the Insulin-Stimulated I_SC_ in Cells Treated with and without 1 µM Aldosterone for 20 h

Figure 2A shows representative responses of I_SC_ to basolateral application of 100 nM insulin in cells with (blue squares in Figure 2A) and without (red circles in Figure 2A) treatment with 1 µM aldosterone for 20 h; insulin was applied at time = 0 in Figure 2A. Insulin induced a biphasic change in I_SC_: an increase followed by a decrease in I_SC_ irrespective of aldosterone treatment (Figure 2A). Aldosterone treatment significantly increased the basal I_SC_ (without insulin application) 9.6-fold to 4.02 ± 0.07 µA/cm^2^ (*n* = 5) from 0.42 ± 0.03 µA/cm^2^ (*n* = 5; *p* < 0.000001). To compare the time-dependent change in the insulin-stimulated I_SC_ in aldosterone-treated cells with that in aldosterone-untreated cells, we normalized the I_SC_ to its peak value in each case (Figure 2B). As shown in Figure 2B, the I_SC_ reached faster its peak value in aldosterone-treated cells than in aldosterone-untreated cells (31.4 ± 1.4 min in aldosterone-treated cells (*n* = 5); 59.0 ± 1.1 min in aldosterone-untreated cells (*n* = 5; *p* < 0.000001)). After reaching its peak value, the insulin-stimulated I_SC_ in aldosterone-treated cells (blue squares and line in Figure 2B) started to decline faster than in aldosterone-untreated cells (red circles and line in Figure 2B), but the insulin-stimulated I_SC_ in aldosterone-treated cells (blue squares and line in Figure 2B) remained at a level almost identical to that in aldosterone-untreated cells (red circles and line in Figure 2B) 5 h after insulin application.

### 2.4. Aldosterone Action on $k_{I}$, $k_{E}$, $k_{R}$ and $k_{D}$ Under the Insulin-Stimulated and -Unstimulated Conditions

Insulin has been reported to stimulate translocation of ENaC to the apical membrane [27]. Apical (t) (see Equation (6)) represents the amount of ENaC in the apical membrane state (Figure 1) at time = t; i.e., I_SC_ changes proportionally to the value of Apical (t) represented by Equation (6), Apical (t). Therefore, considering that the insulin-induced time-dependent change in I_SC_ is due to the change in the amount of ENaC in the apical membrane (Apical (t) represented by Equation (6)), we fitted Equation (6) to the experimentally measured I_SC_, obtaining the values of $k_{I}$, $k_{E}$, $k_{R}$, and $k_{D}$ under the insulin-stimulated condition (Table 1). Aldosterone treatment under the insulin-stimulated condition significantly increased $k_{I}$ 3.3-fold and $k_{R}$ 2.0-fold, but decreased $k_{D}$ to 0.7-fold without showing any significant effect on $k_{E}$. These observations mean that aldosterone treatment elevates the ENaC insertion rate to the apical membrane from the insertion state and the ENaC recycling rate, but diminishes the ENaC degradation rate without showing any effect on the ENaC endocytotic rate under the insulin-stimulated condition (Table 1).

We also studied the effect of aldosterone on $k_{I}$, $k_{E}$, $k_{R}$, and $k_{D}$ under the basal condition without insulin stimulation (Insulin (−) in Table 1). Aldosterone treatment decreased $k_{E}$ 0.5-fold and increased $k_{R}$ 1.4-fold without any significant effect on $k_{I}$ or $k_{D}$. Taken together these observation are as follows: (1) aldosterone increases $k_{I}$ in insulin-stimulated but not -unstimulated cells; (2) aldosterone has no influence on $k_{E}$ in insulin-stimulated cells, but decreases $k_{E}$ in insulin-unstimulated cells; (3) aldosterone increases $k_{R}$ in both insulin-stimulated and -unstimulated cells; (4) aldosterone decreases $k_{D}$ in insulin-stimulated but not -unstimulated cells. This means that the action of aldosterone on the intracellular ENaC trafficking is modified by insulin stimulation: i.e., insulin masks the inhibitory action of aldosterone on $k_{E}$, while insulin induces a stimulatory action of aldosterone on $\;k_{I}$ and an inhibitory action of aldosterone on $k_{D}$, which are not affected by aldosterone alone in the absence of insulin.

### 2.5. Insulin-Induced Time-Dependent Changes in the Amounts of ENaC Localized in Four States, Insert, Apical, Recycl and Degrad, Shown in Figure 1 in Cells Treated with and without Aldosterone (ALDO, 1 µM) for 20 h

We evaluated the amounts of ENaC in the insertion state, the apical membrane state, the recycling state and the degradation state (Figure 3): Insert (t), the amount of ENaC in the insertion state (Insert) at time = t; Apical (t), the amount of ENaC in the apical membrane state (Apical) at time = t; Recycl (t), the amount of ENaC in the recycling state (Recycl) at time = t; Degrad (t), the amount of ENaC in the degradation state (Degrad) at time = t; t is the time after basolateral application of insulin. The solid and dot lines respectively show the amounts of ENaC in cells with and without aldosterone treatment. Figure 3A represents the amount of ENaC in the insertion state; Figure 3B, the amount of ENaC in the apical membrane state; Figure 3C, the amount of ENaC in the recycling state; Figure 3D, the amount of ENaC in the degradation state.

### 2.6. Recycling Ratio of Endocytotic ENaC to the Apical Membrane, $R_{R}$, Under the Insulin-Stimulated and -Unstimulated Conditions

The recycling ratio of ENaC ($R_{R}$) is represented by the following equation:(14)$$R_{R}=\frac{k_{R}}{k_{R}+k_{D}}$$

Aldosterone treatment increased the recycling ratio of ENaC to the insertion state from the recycling state ($R_{R}$) 1.7-fold to 65.86 ± 2.75% from 39.48 ± 6.87% (*n* = 5; *p* < 0.025) in insulin-stimulated cells (*n* = 5; *p* < 0.01; Insulin (+) in Table 2). In insulin-unstimulated cells, aldosterone increased $R_{R}$ 1.3-fold to 41.86 ± 2.39% from 30.83 ± 0.46% (*n* = 4; *p* < 0.025; Insulin (−) in Table 2). Insulin had no significant effect on $R_{R}$ in aldosterone-untreated cells (compare $R_{R}$ in Insulin (+) and (−) in ALDO (−) in Table 2), but significantly increased $R_{R}$ in aldosterone-treated cells (*p* < 0.001; compare $R_{R}$ in Insulin (+) and (−) in ALDO (+) in Table 2). In other words, the stimulatory action of insulin on $R_{R}$ is induced by aldosterone treatment.

### 2.7. Aldosterone Action on Relocation Number of ENaC to the Apical Membrane State, $N_{R}$ (= $k_{R}$/$k_{D}$), Under the Insulin-Stimulated and -Unstimulated Conditions

We also evaluated how many times ($N_{R}$) individual ENaC was translocated to the apical membrane state after the first endocytosis process (Figure 1). $N_{R}$ was evaluated using Equation (15). Aldosterone increased $N_{R}$ 2.7-fold to 2.01 ± 0.26 from 0.74 ± 0.20 (*n* = 5; *p* < 0.005; Table 2) in cells stimulated by insulin, while aldosterone increased $N_{R}$ 1.6-fold to 0.71 ± 0.07 from 0.45 ± 0.01 (*n* = 4; *p* < 0.01) in cells without insulin stimulation (Table 2). These observations suggest that the stimulatory action of aldosterone on $N_{R}$ is enhanced by insulin.
(15)$$N_{R}=\left(\frac{k_{R}}{k_{R}+k_{D}}\right)+{\left(\frac{k_{R}}{k_{R}+k_{D}}\right)}^{2}+{\left(\frac{k_{R}}{k_{R}+k_{D}}\right)}^{3}+\cdots \cdots \cdots =\overset{\infty }{\underset{i=1}{\sum }}{\left(\frac{k_{R}}{k_{R}+k_{D}}\right)}^{i}=\frac{k_{R}}{k_{D}}$$

### 2.8. Aldosterone Action on Cumulative Na^+^ Absorption (I_SC_) ($I_{Total}$) Under the Insulin-Stimulated and -Unstimulated Conditions

We further evaluated the aldosterone action on cumulative Na^+^ absorption ($I_{Total}$) under the insulin-stimulated condition. Aldosterone treatment increased the simulated $I_{Total}$ 3.8-fold to 91,154 ± 2334 μC/cm^2^/day from 23,947 ± 1777 μC/cm^2^/day in insulin-stimulated cells (n = 5; *p* < 0.0001; insulin (+) in Table 3), and 6.2-fold to 50,345 ± 3727 μC/cm^2^/day from 8111 ± 2548 μC/cm^2^/day in insulin-unstimulated cells (n = 4; *p* < 0.0001; insulin (−) in Table 3). Insulin also elevated the simulated cumulative ENaC-mediated epithelial Na^+^ transport irrespective of aldosterone treatment (compare Insulin (+) with Insulin (−) in each case of ALDO (+) or (−); Table 3; *p* < 0.001).

### 2.9. Total Amount of ENaC ($T_{ENaC}$) in Cells with and without Aldosterone Treatment (1 µM for 20 h)

Total amount of ENaC ($T_{ENaC}$) was measured just before insulin application, meaning that the measured $T_{ENaC}$ is not affected by insulin stimulation. In cells treated with aldosterone (ALDO (+)), $T_{ENaC}$ measured in experiments for Insulin (+) was identical to that for Insulin (−) (Table 3). In cells without aldosterone treatment (ALDO (−)), the measured $T_{ENaC}$ for experiments for Insulin (+) was identical to that for Insulin (−) (Table 3). Aldosterone treatment increased $T_{ENaC}$ 2.2-fold in Insulin (+) experiment and 2.0-fold in Insulin (−) (Table 3), while aldosterone treatment increased the amount of ENaC in the apical membrane just before application of insulin (the basal I_SC_) was increased 9.6-fold to 4.02 ± 0.07 (*n* = 5) from 0.42 ± 0.03 in cells used for Insulin (+) experiment (*n* = 5; *p* < 0.0001) and 8.6-fold to 4.02 ± 0. 22 (*n* = 4) from 0.47 ± 0.03 in cells used for Insulin (−) experiment (*n* = 4; *p* < 0.0001). These observations indicate that the aldosterone treatment increased the distribution of ENaC in the apical membrane even considering the elevation of ENaC production.

### 2.10. Aldosterone Action on the Residency Time How Long an Individual ENaC Stays in the Apical Membrane Each Time After the Insertion of ENaC into the Apical Membrane ($T_{AM}=1/k_{E}$) Under the Insulin-Stimulated and -Unstimulated Conditions

As shown in Table 3, the treatment with 1 µM aldosterone for 20 h increased respectively the cumulative Na^+^ absorption 3.8-fold in insulin-stimulated cells and 5.9-fold in insulin-unstimulated cells, while the aldosterone treatment increased $T_{ENaC}$ 2.2-fold in Insulin (+) experiment and 2.0-fold in Insulin (−) (Table 3). This means that the aldosterone-induced increase in the cumulative Na^+^ absorption ($I_{Total}$) would not be only due to the aldosterone-induced increase in ENaC production ($T_{ENaC}$). The cumulative Na^+^ absorption ($I_{Total}$) (Table 3) depends on: (1) the total number of ENaC produced in cells, (2) the residency time of ENaC in the plasma membrane, (3) the single channel conductance of ENaC, (4) the open probability (Po) of ENaC, and (5) the driving force of Na^+^ entry through ENaC into the intracellular space across the apical membrane (the difference between the apical membrane potential and the equivalent potential for Na^+^ across the apical membrane).

First, we studied the effect of aldosterone treatment on the residency time how long an individual ENaC stays in the apical membrane each time after the insertion of ENaC into the apical membrane ($T_{AM}$) under the insulin-stimulated and -unstimulated conditions. Since $k_{E}$ is the endocytotic rate, $1/k_{E}$ means the residency time of ENaC in the apical membrane each time after the insertion of ENaC into the apical membrane (see Equation (16)). Aldosterone treatment had no significant effect on $T_{AM}$ in cells stimulated by insulin (Insulin (+) in Table 4); this phenomenon was expected from the observation that the aldosterone treatment had no significant action on the endocytotic rate of ENaC ($k_{E}$) in cells simulated by insulin (Insulin (+) in Table 1). On the other hand, in cells without insulin stimulation, aldosterone increased $T_{AM}$ 2.4-fold to 0.45 ± 0.08 from 0.19 ± 0.02 (*p* < 0.005; *n* = 4; Insulin (−) in Table 4). This means that insulin masks the aldosterone action on $T_{AM}$.
(16)$$T_{AM}=\frac{1}{k_{E}}$$

### 2.11. Aldosterone Action on the Cumulative Time How Long an Individual ENaC Stays in the Apical Membrane During Its Whole Life-Time Period Before Degradation ($T_{CTAM}=\left(1+K_{R}/k_{D}\right)/k_{E})$ Under the Insulin-Stimulated and -Unstimulated Conditions

We next studied the aldosterone action on the cumulative time $\left(T_{CTAM}\right)$ how long an individual ENaC stayed in the apical membrane before degradation, reflecting the cumulative Na^+^ absorption (see Equation (17) and Table 3):(17)$$T_{CTAM}=\left(1+N_{R}\right)\frac{1}{k_{E}}=\left(1+\frac{k_{R}}{k_{D}}\right)\;\frac{1}{k_{E}}$$

Aldosterone treatment increased $T_{CTAM}$ 1.6-fold to 1.44 ± 0.08 h from 0.88 ± 0.11 h in insulin-stimulated cells (*p* < 0.005; *n* = 5; Insulin (+) in Table 4) and 2.7-fold to 0.75 ± 0.11 h from 0.28 ± 0.02 h (*p* < 0.005; *n* = 4; Insulin (−) in Table 4). We further evaluated the value of $T_{ENaC}$ × $T_{CTAM}$. Aldosterone treatment increased the value of $T_{ENaC}$ × $T_{CTAM}$ 3.7-fold to 25.31 ± 0.65 (*n* = 5) from 6.92 ± 0.33 in insulin-stimulated cells (*p* < 0.0001; *n* = 5), which was almost identical to the aldosterone-induced increase in the cumulative Na^+^ absorption ($I_{Total})$, 3.8-fold, in insulin-stimulated cells (Insulin (+) in Table 3). On the other hand, under the insulin-unstimulated condition, aldosterone treatment increased the value of $T_{ENaC}$ × $T_{CTAM}$ 5.5-fold to 12.83 ± 0.85 (*n* = 4) from 2.32 ± 0.63 (*p* < 0.0001; *n* = 4), which was almost identical to the aldosterone-induced increase in the cumulative Na^+^ absorption ($I_{Total})$, 6.2-fold, in insulin-stimulated cells (Insulin (+) in Table 3).

These results indicate that insulin had additive actions on $I_{Total}$ and $T_{CTAM}$ even in aldosterone-treated cells, and that aldosterone had additive actions on $I_{Total}$ and $T_{CTAM}$ even in insulin-stimulated cells insulin. These observations suggest that aldosterone have additive influence on the intracellular ENaC trafficking, even if insulin and aldosterone have common pathways in modification of the intracellular ENaC trafficking.

The cumulative Na^+^ absorption ($I_{Total}$) (Table 3) depends on: (1) the total number of ENaC produced in cells ($T_{ENaC}$), (2) the residency time of ENaC in the plasma membrane $T_{CTAM}$, (3) the single channel conductance of ENaC, (4) the open probability of ENaC, and (5) the driving force of Na^+^ entry through ENaC into the intracellular space across the apical membrane (the difference between the apical membrane potential and the equivalent potential for Na^+^). The effect of aldosterone treatment on the cumulative Na^+^ absorption in both insulin-stimulated and -unstimulated cells would be quantitatively explained by these two factors, $T_{ENaC}$ and $T_{CTAM}$. Therefore, we suggest that aldosterone would elevate the cumulative Na^+^ absorption ($I_{Total}$) mainly via increases of $T_{ENaC}$ and $T_{CTAM}$ with little effects on the single channel conductance of ENaC, the open probability of ENaC, or the driving force of Na^+^ entry through ENaC into the intracellular space across the apical membrane (the difference between the apical membrane potential and the equivalent potential for Na^+^), although we could not exclude a possibility that the estimated $T_{ENaC}$ might include the aldosterone action on the single channel conductance of ENaC, the open probability of ENaC, and the driving force of Na^+^ entry through ENaC into the intracellular space across the apical membrane.

### 2.12. Aldosterone Action on Whole Life-Time of ENaC after the Initial Insertion to the Apical Membrane; $T_{WLT}$, Under the Insulin-Stimulated and -Unstimulated Conditions

We further studied the aldosterone treatment on the whole lifetime of ENaC ($T_{WLT}$) after the initial insertion of ENaC into the apical membrane under the insulin-stimulated and -unstimulated conditions. The whole lifetime of ENaC ($T_{WLT}$) is described by Equation (18):(18)$$T_{WLT}=\frac{1}{k_{E}}+N_{R}\left(\frac{1}{k_{R}}+\frac{1}{k_{I}}+\frac{1}{k_{E}}\right)+\frac{1}{k_{D}}$$
where $N_{R}$ (Equation (15)) is the number how many times an individual ENaC is recycled to the apical membrane after its first endocytosis from the apical membrane into the recycling state. Aldosterone treatment had no significant effect on the values of $T_{WLT}$ of ENaC after the first insertion into the apical membrane under the insulin-stimulated condition (Table 4), while under the inulin-unstimulated condition, aldosterone significantly increased $T_{WLT}$ 1.5-fold to 4.18 ± 0.41 from 2.82 ± 0.15 (*n* = 4; *p* < 0.025; Table 4). This means that the action of aldosterone on $T_{WLT}$ is masked by insulin.

## 3. Discussion

In the present study, we found that: (1) under the insulin-stimulated condition, aldosterone treatment (1 µM for 20 h) significantly elevated the ENaC insertion to the apical membrane ($k_{I}$) 3.3-fold and the ENaC recycling rate ($k_{R}$) 2.0-fold, but diminished the ENaC degradation rate ($k_{D}$) 0.7-fold without any significant effect on the ENaC endocytotic rate ($k_{E}$); (2) under the insulin-unstimulated condition, aldosterone treatment decreased $k_{E}$ 0.5-fold and increased $k_{R}$ 1.4-fold, without any significant effect on $k_{I}$ or $k_{D}$. Thus, the present study indicates that: (1) insulin masks the well-known inhibitory action of aldosterone on the ENaC endocytotic rate; (2) insulin induces a stimulatory action of aldosterone on ENaC apical insertion and an inhibitory action of aldosterone on ENaC degradation; (3) insulin enhances the aldosterone action on ENaC recycling; and (4) insulin has a more effective action on diminution of ENaC endocytosis than aldosterone.

Aldosterone is well known to diminish the endocytotic rate of ENaC, leading ENaC to stay in the apical membrane for longer time. However, the present study indicates that insulin masks this action of aldosterone on the ENaC endocytotic rate. This observation suggests that insulin has more effective action on the ENaC endocytotic rate than aldosterone. Further, the present study indicates that aldosterone increases the whole-life residency time of ENaC in the apical membrane by elevating the recycling rate associated with diminution of degradation rate in addition to production of ENaC proteins via an increase in ENaC mRNA irrespective of insulin stimulation.

In the present study, we have reported that aldosterone increases apical ENaC expression. This observation is strongly supported by an experimental result reported by Weisz et al. [29]; i.e., the experiment with biotinylation of apical cell surface proteins indicates that 1 µM aldosterone treatment for 18 h in A6 cells increases the apical ENaC expression [29].

Weisz et al. [29] have also reported the insulin action on apical ENaC expression: i.e., application of insulin (100 mU/mL ≈ 600 nM) for 30 min has no significant effect on apical ENaC expression in A6 cells, indicating a contradictory result to that shown in the present study. As one of the reasons causing these contradictory results, we should consider the difference of I_SC_ amplitudes reported in the study by Weisz et al. [29] from that shown in the present study. Weisz et al. [29] have reported that: (1) the basal I_SC_, 4.4 µA/cm^2^, in cells without treatment by aldosterone or insulin, 10-fold larger than that in the present study; (2) the I_SC_, 24.8 µA/cm^2^, in cells treated with 1 µM aldosterone for 18 h, 5-fold larger than that in the present study; and (3) the I_SC_, 12.1 µA/cm^2^, in cells treated with insulin of 100 mU/mL (≈ 600 nM) for 30 min, 10-fold larger than that in the present study. The much larger I_SC_ reported in the study by Weisz et al. [29] would be due to much larger amounts of total and/or apical ENaC expression, compared with those in the present study. If so, the contradictory results regarding the insulin action on intracellular ENaC trafficking would be caused by the different amounts of ENaC expression in the studies by Weisz et al. [29] and the present study, since Weisz et al. [29], and Taruno and Marunaka [30] have reported that the amount of ENaC expression affects intracellular ENaC trafficking.

We also consider another possible explanation on these contradictory results as follows: (1) ENaC has very low and high open probabilities (Po) [31,32]; (2) the method applied in the present study might recognize ENaC with very low Po in the apical membrane state as ENaC in the insert state, but not in the apical membrane state; (3) insulin might increase Po of ENaC with very low Po already localized in the apical membrane state; and (4) the insulin-induced increase in Po of ENaC with very low Po localized in the apical membrane might lead to a conclusion that insulin induces new appearance of ENaC in the apical membrane.

We should further take into consideration another possibility causing the contradictory results. Weisz et al. [29] have reported that application of insulin (100 mU/mL ≈ 600 nM) for 30 min increase I_SC_ 2.8 fold, while insulin (100 nM) has been applied for 5 h in the present study and insulin increases I_SC_ 4.7-fold about 60 min after insulin application. These observations mean that insulin application only for 30 min would not show its maximal effect on I_SC_ (i.e., ENaC), and this short period of insulin application might be a reason why Weisz et al. [29] observe no detectable change in apical ENaC expression. In the insulin-stimulated cells without aldosterone treatment (the same condition as that Weisz et al. [29] used for the study on the insulin action), the time period of ENaC staying in the insertion state ($1/k_{I}$; see $k_{I}$ in Table 1) is 2.33 ± 0.34 h. Using this value of $1/k_{I}$ = 2.33 h, we estimated how many ENaC is trafficked to the apical membrane within insulin application for 30 min, indicating that only 20% ENaC in the insert state is trafficked to the apical membrane for 30 min. If the amount of apical ENaC would be much larger than that of ENaC in the insert state, insertion of only 20% ENaC in the insertion state to the apical membrane would result in little, non-detectable increase in the amount of apical ENaC. Taken together, for the first 30 min of insulin application, insulin would show its stimulatory action on I_SC_ mainly via an increase in Po of ENaC in the apical membrane, but some part of its action would be mediated via an increase in the number of ENaC in the apical membrane, although further studies are required to clarify the contradictory results between observations reported in Weisz et al. [29] and the present study.

Gonzalez-Montelongo et al. [33] have reported the insertion and endocytotic rates of ENaC: the average time of ENaC insertion to the apical membrane from the intracellular store state is ~2 h identical to that reported in the present study ($1/k_{I},\;$= ~2 h; Insulin (+) and ALDO (−) in Table 1); the average time of ENaC in the apical membrane is ~1 h, which is not so far from that reported in the present study ($1/k_{E}$ = ~0.5 h; $T_{AM}=1/k_{E}$ in Insulin (+) and ALDO (−) in Table 4).

Aldosterone has been well known to increase: (1) ENaC expression via elevation of ENaC mRNA expression, and (2) the residency time of ENaC in the apical membrane [34,35]. The latter action of aldosterone on ENaC residency time in the apical membrane is mediated via an increase in expression of serum- and glucocorticoid-induced kinase 1 (SGK1) [34,35], which diminishes endocytosis of ENaC by inhibiting activity of E3 ubiquitin ligase Nedd4-2 via phosphorylation of Nedd4-2 [36]. This means that aldosterone decreases the endocytotic rate of ENaC [22,34,35]. Indeed, the present study indicates that under the insulin-unstimulated condition, aldosterone shows an inhibitory action on ENaC endocytotic rate. On the other hand, under the insulin-stimulated condition, the present study has reported that the aldosterone treatment induces no change in the endocytotic rate of ENaC but increases the insertion rate of ENaC to the apical membrane. The conclusion of the aldosterone action on the insertion and endocytotic rates of ENaC under the insulin-stimulated condition might be a controversial one compared with the previous reports [22,34,35], although the present study has been conducted under a condition stimulated by insulin, which would modulate the action of aldosterone on intracellular trafficking of ENaC.

After endocytosis of ENaC with ubiquitination by Nedd4-2, the ubiquitinated ENaC is trafficked to the lysosome, although some other recycling pathways are also known in the intracellular ENaC trafficking [37]. The detailed information on molecular mechanisms of the intracellular ENaC trafficking has been reviewed in a recently published article [37]. The present study indicates that aldosterone increases the recycling rate of ENaC leading to an increase in the time duration of an individual ENaC staying in the apical membrane during its whole life-time period before degradation ($T_{CTAM}$). There are possible several pathways in intracellular recycling pathways of ENaC [37]. Rab small GTPases are known to regulate ENaC recycling [38]. One of the most important points regarding the regulation of the intracellular ENaC trafficking process is how the ubiquitinated ENaC is determined to be trafficked to the recycling process or the degradation process. The key is deubiquitylation of ENaC performed by deubiquitinase enzymes [37,39]. Aldosterone would increase the recycling rate of ubiquitinated ENaC by deubiquitinating ENaC in not only the apical membrane but also in the lysosome. Therefore, the aldosterone action on the recycling and degradation rates of ENaC shown in the present study would be mediated via an increase of deubiquitinating enzyme activity and/or expression. Further, aldosterone is known to affect activity of small GTPases [40]. Thus, the observation shown in the present study would be also mediated through the aldosterone action on Rab small GTPase.

Insulin is also known to elevate the ENaC-mediated transepithelial Na^+^ transport by activating SGK1 via phosphatidylinositide 3’-kinase (PI3K) [27,41,42]. Blazer-Yost and colleagues have reported that insulin stimulates the insertion of ENaC to the apical membrane from the intracellular store site [43,44]. Insulin also stimulates phosphorylation of SGK1 via a PI3K-mediated pathway, leading to phosphorylation (inactivation) of Nedd4-2 [45]. Thus, insulin inhibits the endocytosis of ENaC from the apical membrane via inactivation (phosphorylation) of Nedd4-2, elevating the number of ENaC in the apical membrane.

These reports mentioned above indicate that both insulin and aldosterone elevate the number of ENaC in the apical membrane. However, it was unclear how aldosterone acts on the intracellular trafficking of ENaC by interacting with insulin. Aldosterone is generally known to increase the number of ENaC in the apical membrane via inhibition of ENaC endocytosis by inhibiting Nedd4-2 activity [46,47], although aldosterone would have stimulatory actions on the insertion of ENaC into the apical membrane [37]. The present study suggests that aldosterone elevates the insertion rate of ENaC to the apical membrane but shows no significant action on ENaC endocytosis under the insulin-stimulated condition. This means that the aldosterone action on ENaC endocytosis would be masked by insulin, which inhibits Nedd4-2 activity via a PI3K-dependent SGK1 pathway [45]. RhoA accelerates the insertion of ENaC to the apical membrane by activating phosphatidylinositol-4-phosphate 5-kinase (PI(4)P5K) via activation of Rho kinase [48,49]. Furthermore, RhoA is also known to be activated by aldosterone [50]. Thus, we suggest that at least under the insulin-stimulated condition, aldosterone mainly shows its action on the insertion of ENaC into the apical membrane, which would be mediated via a RhoA-Rho kinase-PI(4)P5K-dependent pathway.

## 4. Materials and Methods

### 4.1. Chemicals

NCTC-109 medium and fetal bovine serum were obtained from GIBCO (Grand Island, NY, USA). All other chemicals from Sigma (St. Louis, MO, USA.).

### 4.2. Solutions

The bathing solution contained 120 mM NaCl, 3.5 mM KCl, 1 mM CaCl_2_, 1 mM MgCl_2_, 10 mM HEPES, and 5 mM glucose. The pH of both solutions was adjusted to 7.4 with NaOH.

### 4.3. Cell Culture

We obtained A6 cells, renal epithelial cells derived from the kidney of *Xenopus laevis*, from American Type Culture Collection (Rockville, MD, USA.) at passage 68, and were cultured on plastic flasks in a humidified incubator in a culture medium at passages 76–84. The composition of the culture media was 75% (*v*/*v*) NCTC-109 (Sigma-Aldrich, Inc.), 15% (*v*/*v*) distilled water, and 10% (*v*/*v*) fetal bovine serum at 27 °C and 1.0% CO_2_ in air [23,25]. We measured I_SC_ from A6 cells seeded at a density of 5 × 10^4^ cells/well onto tissue culture-treated Transwell filter cups (polycarbonate porous membranes; Costar Corporation, Cambridge, MA, USA) and cultured for 13–15 days forming a monolayer.

### 4.4. Measurements of Short-Circuit Current (I_SC_) in Cells Treated with or without 1 µM Aldosterone for 20 h

We measured I_SC_ from cultured A6 cells monolayer on the Transwell filter cup transferred to a modified Ussing chamber (Jim’s Instrument, Iowa City, IA, USA.) by clamping the transepithelial electrical-potential-difference to 0 mV using a high-impedance millivoltmeter (VCC-600, Physiologic Instrument, San Diego, CA, USA.) [23,51,52,53] in the treatment with or without 1 µM aldosterone for 20 h. In the present study, the transepithelial Na^+^ absorption is represented as a positive current (I_SC_). The bathing solution was stirred by bubbling with 21% O_2_/79% N_2_. During the I_SC_ measuring time period, 5 h, no amino acids or serum was supplied to the bathing solution.

### 4.5. Application of Insulin

Insulin (100 nM) was applied to the basolateral side in Ussing chamber.

### 4.6. Temperature

Since A6 cells are derived from the kidney of *Xenopus laevis* (an amphibian cell line), all experiments were performed at 22~23 °C.

### 4.7. Data Presentation and Statistics

Results shown in Tables are expressed as the mean ± standard error (S.E.). Statistical significance was determined by Student’s *t*-test or ANOVA appropriately, and *p* < 0.05 was considered significant.

## 5. Conclusions

The present study indicates that: (A) under the insulin (100 nM)-stimulated condition, aldosterone treatment (1 µM, 20 h) significantly elevated the ENaC insertion rate to the apical membrane ($k_{I}$) 3.3-fold and the ENaC recycling rate ($k_{R}$) 2.0-fold, but diminished the ENaC degradation rate ($k_{D}$) 0.7-fold without any significant effect on the ENaC endocytotic rate ($k_{E}$) (Figure 4A). (B) under the insulin-unstimulated condition, aldosterone (1 µM, 20 h) treatment significantly decreased the ENaC endocytotic rate ($k_{E}$) 0.5-fold and increased the ENaC recycling rate ($k_{R}$) 1.4-fold without any significant effect on the ENaC apical insertion ($k_{I}$) or degradation rate ($k_{D}$) (Figure 4B); However, further studies are required to confirm the mechanism how aldosterone affects the recycling and degradation rates of ENaC.

## Figures and Tables

**Figure 1 ijms-21-03407-f001:**
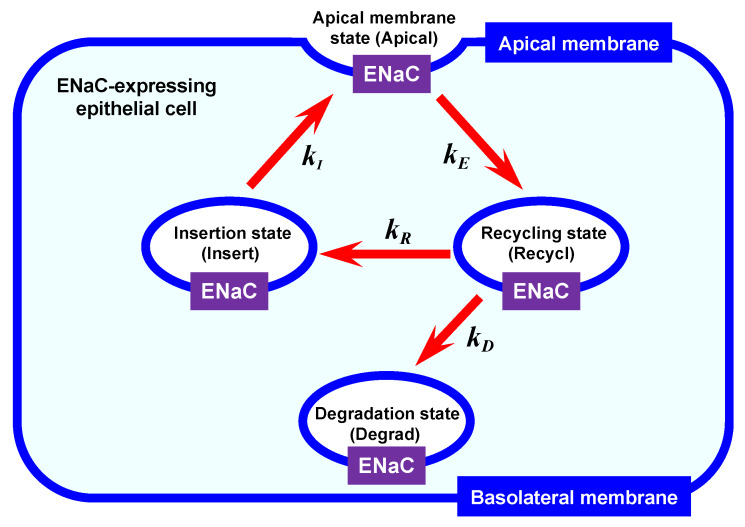
An intracellular ENaC trafficking model. (1) An insertion state (Insert): this state contains epithelial Na^+^ channel (ENaC) that accesses to the apical membrane with an insertion rate into the apical membrane ($k_{I}$). (2) An apical membrane state (Apical): this state contains ENaC that functions as Na^+^-conducting (permeant) pathways across the apical membrane. (3) A recycling state (Recycl): this state contains ENaC retrieved from the apical membrane with an endocytotic rate ($k_{E}$), and then the ENaC is trafficked back to the insertion state (Insert) with a recycling rate ($k_{R}$), or moves to a degradation pathway (Degrad) with a degradation rate ($k_{D}$).

**Figure 2 ijms-21-03407-f002:**
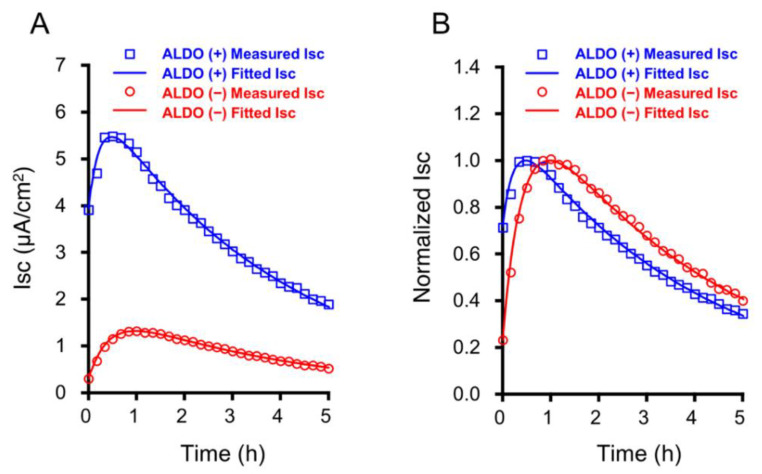
Representative observations of experimentally measured insulin (100 nM)-stimulated short-circuit currents (I_SC_) and simulated I_SC_ using a four-state mathematical model with and without treatment of aldosterone (ALDO, 1 µM for 20 h). (**A**) Blue squares and line respectively indicate a typical time course of experimentally measured insulin-stimulated I_SC_ (blue squares) and a simulated time course of I_SC_ (blue line) in cells with 1 µM aldosterone-treatment for 20 h. Red circles and line respectively indicate a typical time course of experimentally measured insulin-stimulated I_SC_ (red circles) and a simulated time course of I_SC_ (red line) in cells without aldosterone-treatment. (**B**) Normalized I_SC_ to each peak value of I_SC_ = 1. Blue squares and line respectively show the normalized measured I_SC_ and simulated I_SC_ in cells with 1 µM aldosterone-treatment for 20 h. Red circles and line respectively show the normalized measured I_SC_ and simulated I_SC_ in cells without aldosterone-treatment.

**Figure 3 ijms-21-03407-f003:**
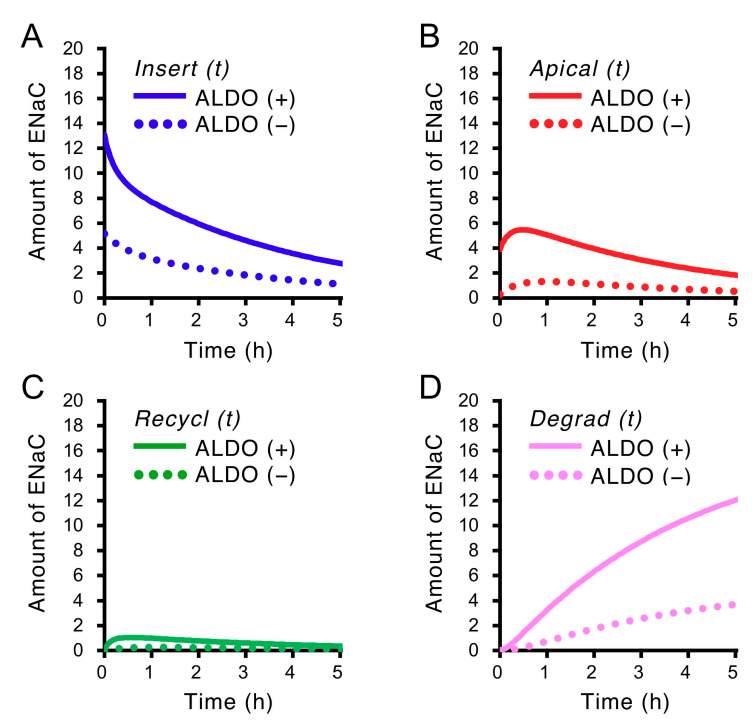
Insulin-induced time-dependent changes in the amounts of ENaC localized in four states, Insert, Apical, Recycl, and Degrad, shown in Figure 1 in cells treated with and without aldosterone (ALDO, 1 µM for 20 h). The amounts of ENaC localized in each state under aldosterone-treated and -untreated conditions are respectively shown by solid (with aldosterone treatment: ALDO (+)) and dot (without aldosterone treatment: ALDO (–)) lines. A, an insertion state (Insert); B, an apical membrane state (Apical); C, a recycling state (Recycl); D, a degradation state (Degrad) as shown in Figure 1. (**A**) Insert (t) (blue lines) shows the amount of ENaC in an insertion state, Insert, at time = t: (**B**) Apical (t) (red lines), the amount of ENaC in an apical membrane state, Apical, at time = t: (**C**) Recycl (t) (green lines), the amount of ENaC in a recycling state, Recycl, at time = t: (**D**) Degrad (t) (pink lines), the amount of ENaC in a degradation state, Degrad, at time = t. t is the time elapsed after application of 100 nM insulin to the basolateral solution. Insert (t), Apical (t), Recycl (t) and Degrad (t) respectively represented by Equations (5)–(8) are described using the values of $k_{I}$, $k_{E}$, $k_{R}$, and $k_{D}$ determined by fitting Apical (t) to the experimentally measured I_SC_.

**Figure 4 ijms-21-03407-f004:**
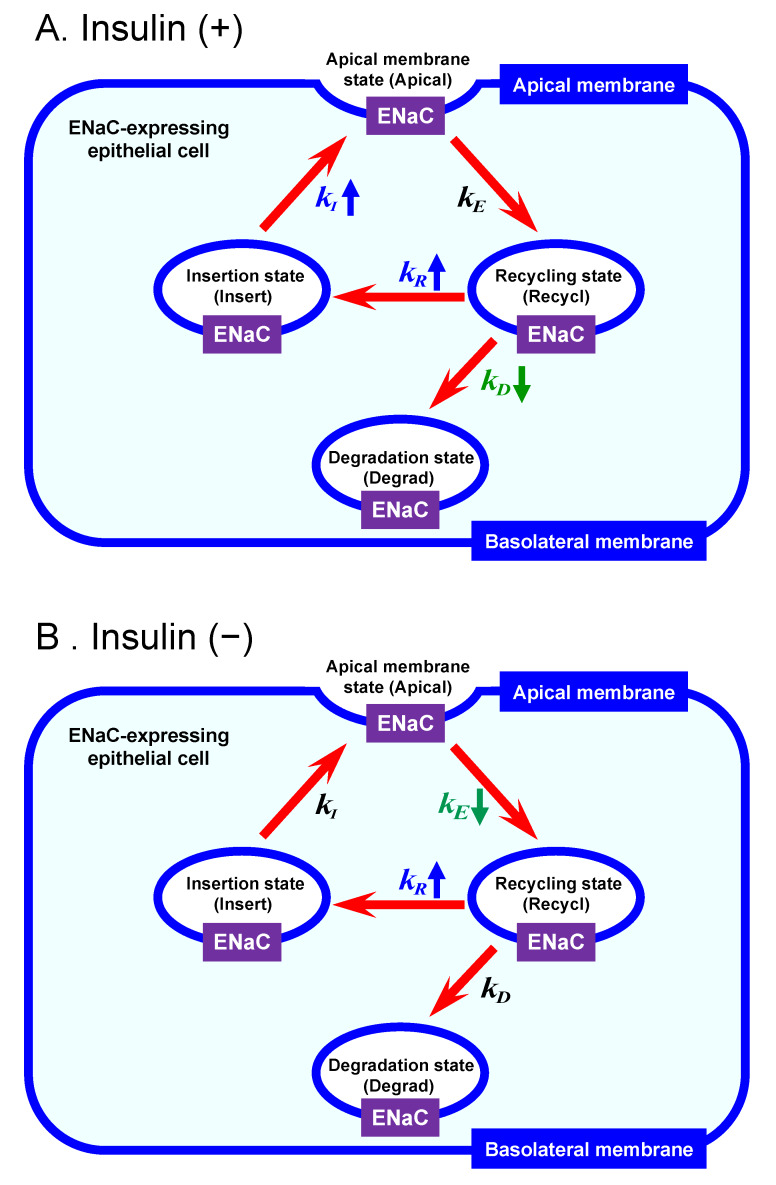
The action of aldosterone treatment (1 µM, 20 h) on the intracellular ENaC trafficking under the condition with (**A**) and without (**B**) stimulation by insulin (100 nM). (**A**) Under the insulin-stimulated condition, aldosterone treatment significantly elevated $k_{I}$ 3.3-fold and $k_{R}$ 2.0-fold, but diminished $k_{D}$ 0.7-fold without any significant effect on $k_{E}$. (**B**) Under the insulin-unstimulated condition, aldosterone treatment significantly decreased $k_{E}$ 0.5-fold and increased $k_{R}$ 1.4-fold without any significant effect on $k_{I}$ or $k_{D}$

**Table 1 ijms-21-03407-t001:** Evaluated values of ENaC’s trafficking rates in the presence and absence of 100 nM insulin with and without 1 µM aldosterone treatment for 20 h.

Insulin	ALDO	Insertion $k_{I}$ (h^−1^)	Endocytosis $k_{E}$ (h^−1^)	Recycling $k_{R}$ (h^−1^)	Degradation $k_{D}$ (h^−1^)
(+)	(+)	1.57 ± 0.06 ^#^	2.12 ± 0.20 ^NS^	6.42 ± 0.99 ^##^	3.16 ± 0.12 ^#^
	(−)	0.47 ± 0.08	1.99 ± 0.74	3.23 ± 0.75	4.59 ± 0.40
(−)	(+)	0.24 ± 0.02 ^NS^	2.45 ± 0.42 *	3.62 ± 0.10 *	5.20 ± 0.46 ^NS^
	(−)	0.21 ± 0.01	5.22 ± 0.45	2.50 ± 0.13	5.62 ± 0.19

ALDO, aldosterone. ^#^ significantly different between ALDO (+) and (−) in Insulin (+) at *p* < 0.01 (*n* = 5). ^##^ significantly different between ALDO (+) and (−) in Insulin (+) at *p* < 0.05 (*n* = 5). * significantly different between ALDO (+) and (−) in Insulin (−) at *p* < 0.005 (*n* = 4). ^NS^ no significant difference between ALDO (+) and (−) in Insulin (+) (*n* = 5), and ALDO (+) and (−) in insulin (−) (*n* = 4). Results shown in Table 1 are expressed as the mean ± standard error (S.E.).

**Table 2 ijms-21-03407-t002:** The recycling ratio, $k_{R}$ (= $k_{R}/\left(k_{R}+k_{D}\right)$) of ENaC and the relocation number how many times ENaC is relocated to the apical membrane state (Apical), $N_{R}$ (= $k_{R}$ /$k_{D}$), after the first retrieval, in the presence and absence of 100 nM insulin with and without 1 µM aldosterone treatment for 20 h.

Insulin	ALDO	Recycling Ratio, $R_{R}$ (= $k_{R}/\left(k_{R}+k_{D}\right)$) (%)	Relocation Number of ENaC to the Apical Membrane State, $N_{R}$ (= $k_{R}$/$k_{D}$)
(+)	(+)	65.86 ± 2.75 ^#^	2.01 ± 0.26 ^#^
	(−)	39.48 ± 6.87	0.74 ± 0.20
(−)	(+)	41.86 ± 2.39 *	0.71 ± 0.07 *
	(−)	30.83 ± 0.46	0.45 ± 0.01

ALDO, aldosterone. ^#^ significantly different between ALDO (+) and (−) in Insulin (+) at *p* < 0.025 (*n* = 5). * significantly different between ALDO (+) and (−) in Insulin (−) at *p* < 0.025 (*n* = 4). Results shown in Table 2 are expressed as the mean ± standard error (S.E.).

**Table 3 ijms-21-03407-t003:** The cumulative Na^+^ absorption (I_SC_) ($I_{Total}$) (μC/cm^2^/day) and the total amount of ENaC ($T_{ENaC}$) under the insulin-stimulated and -unstimulated conditions with and without 1 µM aldosterone treatment for 20 h.

Insulin	ALDO	Cumulative Na^+^ Absorption (I^SC^) ($I_{Total}$) (μC/cm^2^/day)	Total Amount of ENaC ($T_{ENaC}$)
(+)	(+)	91,154 ± 2334 ^#^	7.76 ± 0.82 ^#^
	(−)	23,947 ± 1777	8.21 ± 0.75
(−)	(+)	50,345 ± 3727 *	17.06 ± 0.65 *
	(−)	8111 ± 2548	8.41 ± 0.95

ALDO, aldosterone. ^#^ significantly different between aldosterone (+) and (−) in Insulin (+) at *p* < 0.0001 (*n* = 5). * significantly different between aldosterone (+) and (−) in Insulin (−) at *p* < 0.0001 (*n* = 4). Results shown in Table 3 are expressed as the mean ± standard error (S.E.).

**Table 4 ijms-21-03407-t004:** The residency time of ENaC in the apical membrane how long an individual ENaC stays in the apical membrane each time after the insertion of ENaC into the apical membrane ($T_{AM}=1/k_{E}$) and the cumulative time how long an individual ENaC stays in the apical membrane during its whole life-time period before degradation ($T_{CTAM}=\left(1+K_{R}/k_{D}\right)/k_{E})$, and whole life-time after the first insertion to the apical membrane ($T_{WLT}=1/k_{E}+N_{R}\left(1/k_{R}+1/k_{I}+1/k_{E}\right)+1/k_{D}$) in the presence and absence of 100 nM insulin with and without 1 µM aldosterone treatment for 20 h.

Insulin	ALDO	$T_{AM}$ (h)	$T_{CTAM}$ (h)	$T_{WLT}$ (h)
(+)	(+)	0.49 ± 0.05 ^NS^	1.44 ± 0.08 ^#^	3.35 ± 0.16 ^NS^
	(−)	0.53 ± 0.04	0.88 ± 0.11	2.80 ± 0.32
(−)	(+)	0.45 ± 0.08 *	0.75 ± 0.11 *	4.18 ± 0.41 **
	(−)	0.19 ± 0.02	0.28 ± 0.02	2.82 ± 0.15

ALDO, aldosterone. ^#^ significantly different between aldosterone (+) and (−) in Insulin (+) at *p* < 0.005. (*n* = 5) * significantly different between aldosterone (+) and (−) in Insulin (-) at *p* < 0.005 (*n* = 4). ** significantly different between aldosterone (+) and (−) in Insulin (-) at *p* < 0.025 (*n* = 4). ^NS^ no significant difference between aldosterone (+) and (−) in Insulin (+) (*n* = 5). Results shown in Table 3 are expressed as the mean ± standard error (S.E.).

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
