# Peer review of "Interactive Actions of Aldosterone and Insulin on Epithelial Na+ Channel Trafficking"

_ijms, 2020, doi:10.3390/ijms21103407_

Round 1

Reviewer 1 Report

This is a clever paper that describes the effect of insulin on ENaC trafficking in the presence or absence of aldosterone. The authors use a simple, but complete model.  From this model and from some transepithelial current measurements they calculate several very interesting parameters including the recycling rate and the difference in insertion rate. All of these parameters will be of interest to other investigators in the field.  I have no technical objections. There is one minor issue of English usage. In many places in the manuscript the authors use the phrase "...ENaC staying in the... [some state]". The equations describe the amount in a particular state; therefore, the expression should be "...ENaC in the [some state]". This needs to be corrected throughout the manuscript (it is present in many places).

Author Response

Reviewer’s comments:

There is one minor issue of English usage. In many places in the manuscript the authors use the phrase "...ENaC staying in the... [some state]". The equations describe the amount in a particular state; therefore, the expression should be "...ENaC in the [some state]". This needs to be corrected throughout the manuscript (it is present in many places).

Response:

I revised these words according to Reviewer’s suggestion in the whole manuscript.

I also checked all typos and corrected them in the whole manuscript.

The parts revised in the present version are written in RED.

Reviewer 2 Report

The present study investigates the effect of aldosterone in the cellular trafficking and plasma membrane insertion of ENaC channels. Insulin and aldosterone are well known modulators of intracellular trafficking of ENaC. Through a mathematical modelling of the determination of insulin-stimulated short circuit currents (ISC) in the absence and presence of aldosterone, the authors describe that aldosterone enhances the ENaC insertion in the apical membrane. The issue investigated is interesting. This study is a continuation, and resembles, a recent paper investigating the role of tyrosine kinase inhibitors in ENaC intracellular trafficking.

Major points:

While the study is focused on the effect of aldosterone in insulin-stimulated trafficking of ENaC, in order to support the authors conclusion, it would be important to describe the role of aldosterone alone in the intracellular trafficking of ENaC.

To further support the authors conclusion the study should include a confirmation of the plasma membrane expression of ENaC by biotinylation or immunocytochemistry under the different experimental conditions.

Author Response

he part revised in the present version are written in Red.

Reviewer’s comments:

While the study is focused on the effect of aldosterone in insulin-stimulated trafficking of ENaC, in order to support the authors conclusion, it would be important to describe the role of aldosterone alone in the intracellular trafficking of ENaC.

Response:

We added some experimental results regarding the effect of aldosterone alone on the intracellular trafficking of ENaC according to the comment by Reviewer in Abstract and Results.

See Abstract, Tables 1 ~ 4 and Figure 4 with some description on Tables 1 ~ 4 and Figure 4.

------------------

Reviewer’s comments:

To further support the authors conclusion the study should include a confirmation of the plasma membrane expression of ENaC by biotinylation or immunocytochemistry under the different experimental conditions.

Response:

We added some results according to the comment by Reviewer.

See Lines 337-355.

I also checked all typos and corrected them in the whole manuscript.

Round 2

Reviewer 2 Report

The authors have made an effort to improve the manuscript but some concerns have not been adequately addressed, especially that concerning my previous comment: To further support the authors conclusion the study should include a confirmation of the plasma membrane expression of ENaC by biotinylation or immunocytochemistry under the different experimental conditions. The author refer to three published manuscripts: Weisz et al describe that aldosterone increase the ENaC expression in the plasma membrane but that insulin does not significantly modify its expression, which does not confirm the authors´ hypothesis. Hill et al, explore the expression of ENaC in lipid rafts and this study does not address my concern and the manuscript by Kamynina et al is a review paper. Therefore, the question remains, in addition, and taking into account that Weisz and coworkers have describe that insulin does not alter the plasma membrane expression of ENaC, the authors should provide an explanation for their contradictory results.

Author Response

The revised parts are written in Red.

We added the description. See lines 395-438.

We added possible explanations on the contradictory results reported by Weisz et al and us. See lines 399-438.

We removed tow references, Hill et al. and Kamynina from this manuscript.

Round 3

Reviewer 2 Report

The authors have clarified controversial results with a previously published paper.